# Investigation of the Antitumor Effects of Tamoxifen and Its Ferrocene-Linked Derivatives on Pancreatic and Breast Cancer Cell Lines

**DOI:** 10.3390/ph15030314

**Published:** 2022-03-05

**Authors:** Márton Kalabay, Zsófia Szász, Orsolya Láng, Eszter Lajkó, Éva Pállinger, Cintia Duró, Tamás Jernei, Antal Csámpai, Angéla Takács, László Kőhidai

**Affiliations:** 1Department of Genetics, Cell and Immunobiology, Faculty of Medicine, Semmelweis University, 1085 Budapest, Hungary; martonkalabay@gmail.com (M.K.); szaszzsoccii@gmail.com (Z.S.); langorsi@gmail.com (O.L.); lajesz@gmail.com (E.L.); eva.pallinger@gmail.com (É.P.); angela.takacs1@gmail.com (A.T.); 2Department of Inorganic Chemistry, Faculty of Chemistry, Eötvös Loránd University, 1053 Budapest, Hungary; cinti1994@gmail.com (C.D.); jernei91@gmail.com (T.J.); csampai@caesar.elte.hu (A.C.)

**Keywords:** tamoxifen, GPER1, ferrocene, oxidative stress, cytotoxicity

## Abstract

Tamoxifen is a long-known anti-tumor drug, which is the gold standard therapy in estrogen receptor (ER) positive breast cancer patients. According to previous studies, the conjugation of the original tamoxifen molecule with different functional groups can significantly improve its antitumor effect. The purpose of this research was to uncover the molecular mechanisms behind the cytotoxicity of different ferrocene-linked tamoxifen derivates. Tamoxifen and its ferrocene-linked derivatives, T5 and T15 were tested in PANC1, MCF7, and MDA-MB-231 cells, where the incorporation of the ferrocene group improved the cytotoxicity on all cell lines. PANC1, MCF7, and MDA-MB-231 express ERα and GPER1 (G-protein coupled ER 1). However, ERβ is only expressed by MCF7 and MDA-MB-231 cells. Tamoxifen is a known agonist of GPER1, a receptor that can promote tumor progression. Analysis of the protein expression profile showed that while being cytotoxic, tamoxifen elevated the levels of different tumor growth-promoting factors (e.g., Bcl-XL, Survivin, EGFR, Cathepsins, chemokines). On the other hand, the ferrocene-linked derivates were able to lower these proteins. Further analysis showed that the ferrocene-linked derivatives significantly elevated the cellular oxidative stress compared to tamoxifen treatment. In conclusion, we were able to find two molecules possessing better cytotoxicity compared to their unmodified parent molecule while also being able to counter the negative effects of the presence of the GPER1 through the ER-independent mechanism of oxidative stress induction.

## 1. Introduction

Tamoxifen was first described by Dora Richardson et al. in 1962 as a contraceptive pill called ICI 46,474. A decade later, Craig Jordan reinvented the compound from a failed anticoncipient to the first targeted anti-tumor therapy [1]. It is estimated that hundreds of thousands of women are alive as a result of tamoxifen therapy, and millions more have benefited from it as palliative care or as a chemopreventive approach against cancer [2]. As of today, tamoxifen has become the gold standard therapy of estrogen receptor (ER)-positive breast cancer. However, it has several possible fields of use not related to tumor therapy. It is a generically available, cost-effective, orally administered drug that was taken onto the List of Essential Medicines by the World Health Organization, a list consisting of the most effective and safest medicines needed in a health system.

In today’s era of targeted tumor therapy, tamoxifen was put into a new perspective, and further research has begun with the excessive knowledge about the properties of this compound in hand. The general idea behind new research is either to find a new group of diseases where tamoxifen can be taken into consideration as a treatment option or to modify the original molecule to find a better therapeutic option than the now existing drug.

It became clear in the last two decades that tamoxifen is not only a selective estrogen receptor modulator (SERM), i.e., a drug with ER agonistic or antagonistic properties in a tissue-selective context, but it has several, off-target effects unrelated to the ER. These effects include the cell type and isoform-dependent inhibition of protein kinase C (PKC), which is a key hub of intracellular signal-transducing networks, but tamoxifen also has a general proapoptotic and antiproliferative effect while reducing the migratory activity, neovascularization, and multidrug resistance of tumor cells [3]. These findings helped physicians to experimentally implement tamoxifen in the treatment of several tumor types, such as pancreatic cancer, but also non-oncological diseases, such as hypertrophic cardiomyopathy, mania, Parkinson’s disease, male and female infertility, gynecomastia, retroperitoneal fibrosis, idiopathic sclerosing mesenteritis, and various infections [3,4,5,6,7,8].

In parallel with the promising results with the unmodified tamoxifen in the treatment of a wide spectrum of diseases, the research for a new, modified derivative of tamoxifen that can surpass its parent molecule in effectivity began. The pharmacophore base, which is responsible for most of the effects, can be attached with functional groups that contain, e.g., a halogen atom, a ferrocene group, or a carbo- or heterocycle. Halogenation increases the lipophilic character, and the intracellular penetration of the drug [9], ferrocene-linking has a greatly beneficial redox property [10], while carbo- and heterocycles serve as a base for further modifications that can make the derivative more selective for certain ER isoforms [11]. While each modification has its benefits, the ferrocene-linked derivatives seem to be the most promising for additional research. In Table 1, we collected the most recent results of different authors working with different ferrocene-linked tamoxifen derivatives to compare with our results discussed in this article.

## 2. Results

### 2.1. Cytotoxicity

The cytotoxic effect elicited by the original, unmodified tamoxifen parent molecule and its ferrocene-linked derivatives, T5 and T15, were measured in two steps. These molecules are part of the panel of 20 different derivatives of tamoxifen, which are not discussed in this article. At first, a screening step was conducted to observe whether these compounds possess any kind of cytotoxic activity, and after that, the IC_50_ values were determined in case the molecule was effective on the model cells. 

For the screening step, cells were treated with 10 μM and 100 μM of the selected drug. Viability was measured after 72 and 96 h with the xCELLigence system and the AlamarBlue assay depending on the cell line. A molecule was seen to be fit for further research if the viability decreased to a level lower than 20% after 72 h in any of the two concentrations. Tamoxifen, T5, and T15 were all able to elicit the desired decrease in the viability of each cell line in higher concentrations. 

As a next step, the IC_50_ values of all compounds were measured with the previously used cytotoxicity assays (Table 2). The unmodified tamoxifen proved to be cytotoxic on all three cell lines. Interestingly, the best IC_50_ after 72 h was measured on the MDA-MB-231 cell line (21.8 μM—Table 1), which is considered as an ER-negative tumor type, thus it is outside the classical indications of the compound. The IC_50_ value at 72 h on PANC1 cells (33.8 μM—Table 1) was slightly higher, while MCF7 cells proved to be the least responsive to tamoxifen treatment in our experimental setup.

In comparison with the unmodified parent molecule, the incorporation of a ferrocene group improves the anti-tumor effect of tamoxifen.

The flexible, ferrocene moiety-containing T5 showed higher cytotoxicity on PANC1 cells (IC_50_ at 72 h: 12.5 μM—Table 1) compared to the unmodified parent molecule, however little to no difference was observed on MCF7 (IC_50_: 43.3 μM at 72 h—Table 1) and MDA-MB-231 cells (IC_50_: 26.3 μM at 72 h—Table 1).

The rigid-structured, ferrocene-linked derivative T15 had the best IC_50_ value at 72 h on PANC1 cells (15.0 μM—Table 1), followed by MCF7 (23.0 μM—Table 1) and MDA-MB-231 (23.7 μM—Table 1). After approximately 20–24 h upon treatment with ferrocene-linked derivatives, the viability of cells reached its near achievable minimum, except the case of T15 on PANC1 cells, as the compound was able to lower the viability slightly further after 72 and even 96 h (24 h: 24.4 μM, 72 h: 15.0 μM, 96 h: 5.6 μM). For the IC_50_ curves of each tested molecule, please refer to the Appendix A.

### 2.2. Cell Cycle Analysis

To better understand the mechanism of the anti-tumor effect elicited by our compounds and consequently explain the difference in their activity, we determined their effect on the cell cycle progression. Cells stained with propidium-iodide were studied at 24 and 48 h upon treatment with 25 μM of the compound, as this is the EC_80_ value of tamoxifen for PANC1 cells. Based on clinical experience, PANC1 cells should be the least susceptible to tamoxifen, and the concentration at the EC_80_ value should be low enough not to be immediately toxic to cells. 

All investigated compounds acted in a time-dependent manner. Tamoxifen (Figure 1) was slightly cytotoxic and raised the cell number in the subG1 subpopulation after 24 on PANC1 cells (Figure 1B). On MCF7 cells, an S phase arrest was observed after 24 h following treatment (Figure 1A), yet after 48 h, tamoxifen elicited a directly cytotoxic effect with a remarkable higher subG1 subpopulation (Figure 1D). On MDA-MB-231 cells, tamoxifen treatment resulted in a G1 phase arrest, both after 24- (Figure 2C) and 48-h (Figure 2E) following treatment. On MDA-MB-231 cells, tamoxifen treatment could induce blocking in the G1 phase since the cells were significantly accumulated in of cells in this phase. 

On PANC1 cells, incorporation of a ferrocene group into the parent molecule made no significant impact on the cell cycle after 24 h but elicited a cytotoxic result after 48 h (Figure 2F and Figure 3F). On MCF7 cells, treatment with T15 resulted in a G1 phase arrest in 24 h (Figure 3A) and showed direct cytotoxicity after 48 h (Figure 3D). On MDA-MB-231 cells, T15 showed G1 phase blocking effects in both observed time intervals (Figure 3B,E). T5 treatment made no significant impact on the cell cycle of MDA-MB-231 cells in the first 24 h (Figure 2A,B), but it was able to cause an arrest in the G1 phase in the 48 h time-period (Figure 2D,E) in case of MCF7 and MDA-MB-231 cells. 

### 2.3. ROS Production

There was no significant difference between tamoxifen treatment and the control after 24 h in the cellular oxidative stress on any of the investigated cell lines (Figure 4A–C). The incorporation of a ferrocene group resulted in a significantly elevated ROS generation after 24 h on all cell lines (Figure 5D,E). Treatment with the highest concentration (25 µM) always caused significantly higher oxidative stress than the medium control.

### 2.4. Proteome Profiler

The Proteome Profiler Human XL Oncology Array (R&D Systems, Minneapolis, MN, USA) is a membrane-based, Western blot-like method designed to parallelly measure the levels of 84 different cancer-related proteins, which are important biomarkers of different tumor types involved in several key intracellular pathways. Based on the effectivity of the ferrocene-linked compounds on our model cells, T15 was chosen for proteome analysis, as it produced significantly better IC_50_ values on the MCF7 cell line, which represents the tumor type of the main indication for tamoxifen therapy.

The expression profile induced by T15 was compared to that of the tamoxifen and DMSO. (There was a significant difference between the cell culture medium and DMSO treatment in the expression of several proteins.) Photos of each Proteome Profiler membrane are provided in the Appendix A.

First measurements were conducted on MCF7 cells. The normalized pixel densities of 8 proteins were above the desired 20% of the reference spots. Expression levels of survivin and p53 were additionally included, as they are valuable prognostic markers in breast cancer [23,24], although their normalized pixel density did not reach the desired level. In addition, ER α was only detectable on this cell line. However, its expression did not reach the desirable level (Figure 6A). 

Tamoxifen treatment elevated the expression of amphiregulin, cathepsin D, ErbB3, and survivin compared to both the DMSO control and T15. On the other hand, tamoxifen significantly lowered the levels of p53 compared to both DMSO and T15. T15 treatment was able to lower the levels of amphiregulin, ERα, and survivin, and it also resulted in higher galectin expression compared to tamoxifen. In addition, neither of the compounds had a significant effect on the expression of epithelial cell adhesion protein (EpCAM) and galectin 3 compared to the DMSO control (Figure 6A)

On MDA-MB-231 cells 6 proteins reached the target percentage of the reference spots, and survivin was also included despite its lower expression level (Figure 6B).

Expression of cathepsin D, interleukin 8 (IL8), dickkopf 1 (Dkk1), interleukin 6 (IL6) was significantly raised by tamoxifen treatment compared to the DMSO control and the T15-treated samples. Galectin 3, p53, and survivin were found to be significantly less expressed after treatment with tamoxifen compared to the DMSO control. Galectin 3 level was significantly lower in comparison to tamoxifen after T15 treatment, yet there was no difference in the effect of the two compounds in Dkk1 and p53 levels. Both tamoxifen and T15 were able to decrease survivin expression to an undetectable level (Figure 6B).

PANC1 cells had the most proteins with high expression levels, including key proteins p53 and Bcl-XL, as their normalized pixel intensity was at the desired percentage (Figure 6C).

Amphiregulin, B-cell lymphoma XL (Bcl-XL), IL8, and galectin 3 levels were significantly raised upon tamoxifen treatment compared to both the DMSO control and T15. Dkk1 expression was significantly higher after tamoxifen treatment, yet T15 treatment showed no difference in comparison to the DMSO control. Levels of granulocyte-monocyte colony-stimulating factor (GM-CSF) were significantly increased by T15 in comparison to both DMSO and tamoxifen. Cathepsin S, Osteopontin, Serpin B5, and Serpin E1 expression were upregulated after treatment with the two compounds compared to DMSO control. However, there was no significant difference between their effects. Expression of epithelial growth factor (EGFR), EpCAM, and basic fibroblast growth factor (bFGF) was not influenced either by tamoxifen or T15 compared to the control (Figure 6C). 

### 2.5. Estrogen Receptor Expression Profile

All three cell lines express the G-protein coupled estrogen receptor 1 (GPER1) (Figure 6G–I). The highest intensity was detectable in the permeabilized samples, implying that more receptors were found intracellularly. The highest receptor density was found on MDA-MB-231 cells (Figure 7), followed by PANC1 and MCF7 cells. Merged pictures containing data from all three channels implied that GPER1 was expressed intracellularly in a perinuclear localization, which befits the endoplasmic reticulum (Figure 7G–I). Comparison of lysated and non-lysated samples is provided in the Appendix A.

Although ERα and ERβ were well-characterized ER isoforms, the results in literature findings of their expression on pancreatic cancer cell lines were conflicting [25,26,27]. According to our experimental results, all three cell lines expressed the ERα to a varying degree in nuclear localization (Figure 7A–C). However, only MCF7 and MDA-MB-231 expressed the ERβ (Figure 7D–F).

## 3. Discussion

Cytotoxicity measurements revealed that the addition of ferrocene moiety increases the cytotoxic effect on all investigated cell lines. For a brief outlook, we compared our IC_50_ with that of blood samples of patients undergoing tamoxifen treatment, and the IC_50_ values of all three compounds correlated with the in vivo concentrations of tamoxifen-treated breast cancer patients [28]. 

The background mechanism of this favorable anti-tumor effect can be approached from two sides. It was previously demonstrated that not only tamoxifen, but its ferrocene-linked derivatives are also able to bind to the ER isoforms [9,10,11,29]. Therefore, on ER-expressing cell lines, ER-related target genes might play a role in their anti-tumor actions. However, it is also well known that tamoxifen is capable of interacting with several ER-independent pathways as well. To further elaborate, whether the ER-dependent or ER-independent route is favored by the compounds of this study, after the proper characterization of the estrogen receptor profile of our model cells, we tested their effect on the cell cycle, their impact on the oxidative stress, and also measured whether they altered the expression of key regulators in tumor biology. 

ERα is the most-known nuclear ER isoform, and pathological diagnostics still determine the ER positivity of the tumor based on the presence of the ERα. ERβ, another nuclear ER isoform, is the most expressed in the breast, endometrium and has far less importance in the physiological proliferation of these tissues than the α. There is a vast amount of data about the expression of the β isoform, yet its significance in breast cancer development and progression is not clear [30]. According to our results, all three cell lines express the ERα, but only MCF7 and MDA-MB-231 express ERβ. G-protein-coupled Estrogen Receptor 1 (GPER1) is a third known isoform of estrogen receptors. It is generally different from the other two, as it is a G_s_-coupled seven-transmembrane receptor. The localization of GPER1 is not yet clear, and there is conflicting data, whether it is expressed in the plasma membrane [31] or the endoplasmic reticule [32]. Our results suggest that GPER1 is mainly localized in the endoplasmic reticule and not in the plasma membrane. The highest GPER1 expression was measurable on MCF7 cells, followed by PANC1 and, lastly, MDA-MB-231, which correlates with findings of other authors [33,34]. Tamoxifen belongs to the group of selective estrogen modulators (SERMs), which act as agonists or antagonists on the ER, depending on the tissue type, but several authors suggest that the activation-inhibition profile also depends on the receptor isoform and their ratio in a cell [30,35]. In the breast and pancreas tissues, tamoxifen is a well-known antagonist of the estrogen receptor α and β, but it is an agonist on the GPER1 [36,37,38].

First, cell cycle analysis was conducted to see whether the compounds favor a direct cytotoxic effect or rather cause an arrest in the cell cycle. We demonstrated that both T5 and T15 could arrest the cell cycle in the G1 phase in the breast cancer cells. However, we theorize that this effect might be concentration-dependent. For the cell cycle measurements, 25 μM concentrations of the individual compounds were used. This concentration was chosen at the very beginning of our experimental series because this is the EC_80_ value of tamoxifen on PANC1 cells, the cell line where tamoxifen is theoretically the least efficient as seen in everyday clinical practice. The lowest effective concentration was needed because high concentrations of tamoxifen were described to elicit direct cytotoxicity without a chance to observe its effect on the cell cycle [39]. Comparison of our IC_50_ values of each derivative and the results of the cell cycle analysis revealed that where the IC_50_ value was higher than the 25 μM concentration used for treatment, the derivative was able to cause an arrest in the cell cycle. On the other hand, where the IC_50_ was near or below 25 μM, the derivative elicited a direct cytotoxic effect, raising the cell count in the subG1 population.

The role of the nuclear ER isoforms does not generally differ in breast and pancreatic cancers [40,41]. They can affect the gene expression as nuclear receptors, formation of Erα–ERα, or Erβ–ERβ homodimers upregulate the estrogen-dependent gene expression. However, ERα can form heterodimers with ERβ, in which the ERβ component can disrupt the effect of the ERα on its target genes [35]. Studies suggest that both ERα and ERβ are interacting several extranuclear targets, thus altering different cellular pathways, such as the cell cycle [42] and the PPAR pathway [43]. Estrogen generally acts towards the progression of the cell cycle, either through regulating the expression of different cyclins, cyclin-dependent kinases, and their regulatory proteins or through the activation of the MAPK pathway. Estrogen increases the expression of cyclin D1 and c-myc, while depleting the CDK inhibitor p21^WAF/CIP1^ through the ERα. Cyclin D1 binds to cyclin-dependent kinase 4/6 (CDK4/6), which can activate the complex of cyclin E and CDK2, resulting in the phosphorylation of the retinoblastoma protein (pRB) and the progression from G1 phase to S phase [44]. Elevated cyclin D1 expression alone is not sufficient to activate the cyclin E-CDK2 complex. The ERα-induced expression of c-myc and also the estrogen-induced increase of both cyclin E and CDK2 are required for this process [45]. The transition from the S phase to the G2/M phase is also regulated by the ERα and ERβ since cyclin A, and CDK 2 expression is increased upon estrogen exposure. 

During the synthesis of the T5 and T15 derivatives, the pharmacophore frame of the original tamoxifen was kept as a base, which means that they retained the SERM-like property of tamoxifen, and they similarly interact with them as the parent molecule. Based on these findings, the observed arrest in the cell cycle phases induced by the tamoxifen and its ferrocene-containing derivatives can be explained by the inhibition of ERα and ERβ-regulated gene expression.

Second, the effect on the intracellular oxidative stress upon treatment with the compounds was measured. One of the most prominent advantages of T5 and T15 compared to tamoxifen lies in their redox properties. Ferrocene-containing drugs generally increase the ROS production in treated cells, while they also act as a redox activator and increase the lipophilic character of the drug molecule. Interestingly, there was a significant difference in PANC1 cells in terms of oxidative stress between the flexible and rigid structures of the two derivatives, as the rigid T15 elicited a higher ROS production rate. These characteristics are totally independent of the estrogen receptor expression profile. First, the pharmacophore frame undergoes 4′ hydroxylation, then the ferrocene group can act as an electron relay, while the 4-OH-tamoxifen part of the molecule loses 2 electrons and 2 protons and shifts into a quinone-methide, a more stable intermediate [46]. Quinone-methides are Michael acceptors and are susceptible to attacks from endogenous nucleophiles, such as nucleobases or glutathione, which leads to an increase in ROS production [47]. Quinone-methides are also known to interact with thioredoxin reductases if there is an accessible selenol group in the active site of the enzyme. As thioredoxin reductases are flavoproteins that are involved in the formation of reduced disulphide bonds, interference with this enzyme system might lead to the accumulation of misfolded proteins and consequent apoptosis but it also contributes to the elevated oxidative stress [48]. Hydrogen peroxide was accumulated in all our investigated cell lines in a concentration-dependent manner. This phenomenon is due to the elevated superoxide dismutase activity that transforms superoxide ions generated by the mitochondria and the decreased levels of catalase and glutathione peroxidase. Hydrogen peroxide might play the role of a prodrug. Prodrugs seem to be a beneficial approach to treatment as they are theoretically able to reduce side effects and increase the selectivity of the drug towards the tumor. Under the effect of ferrocene, hydrogen peroxide splits into hydroxyl anions and highly reactive hydroxyl radicals [49].

Linking the original tamoxifen molecule with a ferrocene group can not only lead to an elevated cytotoxic effect but there is a beneficial possibility that it might also make the compound more selective towards the cancerous cell while leaving the non-neoplastic cells intact [50]. Another great advantage of ferrocene-linked derivatives is that they can induce cell death through routes independent from the classic proapoptotic pathway, which bears a vast significance when treatment of apoptosis-resistant tumor types is needed [51]. 

Third, the assessment of the protein expression profile gave us insight into what general pathways play a role in the signal transduction of T15. The investigated 32 proteins regulate cell growth, apoptosis, cell migration, and metastasis development. 

Amphiregulin is an activating ligand of the EGFR, acting towards increased cell growth, and its expression is considered to be a poor prognostic factor. Co-expression of Amphiregulin with EGFR, as seen on PANC1 cells, results in an autocrine feedback loop that associates with poor tumor differentiation [52]. Both MCF7 and PANC1 cells sustained their Amphiregulin levels after T15 treatment compared to DMSO, while tamoxifen increased the levels of this ligand. ErbB3 is also a key player in the EGFR signaling route, as it is a preferred dimerization partner of EGFR. In vivo studies suggest that the high ErbB3 expression correlates with a larger tumor volume and might lead to the failure of both hormonal and tyrosine kinase inhibitor-based anti-tumor therapies.

P53 and Bcl-XL are two key regulatory proteins of the apoptosis pathway. P53 positivity correlates with worse initial prognosis and disease recurrence in ER-negative and positive breast cancer [53,54]. In pancreatic cancer, high p53 levels are signs of sensitivity to therapy, but it is also a bad prognostic factor if it is observed post-therapy [55]. Bcl-XL expression was detected only on PANC1 cells. Its key roles are the promotion of cell survival and evasion of apoptosis [56]. T15 treatment proved to be non-inferior to tamoxifen regarding its effects on the p53 expression. Survivin belongs to the family of Inhibitors of Apoptosis (IAP) proteins and plays a role in angiogenesis and cell proliferation control. Consequently, higher survivin expression is identified as a worse prognostic factor in breast cancer [24] and pancreatic cancer [57]. T15 was able to decrease survivin expression to a non-detectable level in all observed cases. 

Levels of different cytokines were elevated after treatment with both tamoxifen and T15 on both MDA-MB-231 and PANC1 cells. IL8 is constitutively expressed by pancreatic and breast cancers, and its main function is the enhancement of angiogenesis. It can also serve as an autocrine growth factor on PANC1 and MDA-MB-231 cells, as they both express the Interleukin 8 Receptor (IL8R) [58,59]. Chemokine (C-C motif) ligand 20 (CCL20) only was expressed in PANC1 cells, and it promotes metastasis development in vivo [60]. IL6 expressed in MDA-MB-231 cells is associated with a chronic inflammatory tumor microenvironment and resistance to therapy [61]. Both CCL20 and IL8 might lead to resistance to therapy, as they can be overexpressed as a compensatory mechanism during anticancer treatment [60,62,63]. The advantage of T15 is that although it elevates the aforementioned cytokine levels, it is not as high as tamoxifen treatment does. 

Several tumor-invasiveness-related proteins were detected on our cell lines. Galectin 3, a β-galactoside-binding lectin, is a regulator of cell-cell and cell-matrix interactions [64]. EpCAM functions as an epithelium-specific intercellular adhesion molecule that can also initiate intracellular signaling, migration, differentiation, and proliferation [65]. Dkk1 is a spatially and timely expressed antagonist of the Wnt/β-catenin pathway, which is a key player in tumor progression [66]. High expression levels of Galectin 3 and EpCAM are associated with poor patient survival rates and a higher metastatic disease burden in all tumor types of this study. According to our results, Galectin 3 was detected on all three cell lines. T15 treatment was the most beneficial on MDA-MB-231 cells in the aspect of this protein, as it was able to repress Galectin 3 expression to lower levels than tamoxifen. Regarding EpCAM, the ferrocene-linked T15 molecule did not elicit a better cell biological response compared to tamoxifen. Depending on the tumor type, Dkk1 upregulation has a conflicting roleDkk1 overexpression is involved in the invasiveness of pancreatic ductal adenocarcinoma [67], while in breast cancer, it is responsible for the lower proliferation ability via keeping the Wnt/β-catenin pathway under control [68]. T15 treatment did not increase the Dkk1 expression compared to the control on PANC1 cells, while tamoxifen did, thus the effect of T15 was more favorable regarding this protein. Cathepsins are a family of cysteine proteases, and their overexpression of several subtypes is shown in different human cancers with a metastatic phenotype, however, different cathepsins might have different roles depending on the tumor type. Increased Cathepsin B and D levels correlate with poor disease-free survival in breast cancer patients, independent of the hormone receptor status of the tumor [69], and also in pancreatic cancer patients [70]. On the other hand, high Cathepsin S expression on tumor cells associate with better TNM status and a higher survival rate of patients [71]. Treatment with T15 demonstrated elevated levels of Cathepsin S compared to both tamoxifen and control on PANC1 cells, while on the two breast cancer cells, a change in the ratio of different cathepsin isoforms was observed upon the treatment, which is no clear predictor of the invasiveness of the tumor.

Proteins of our screening assay cover a wide range of functions, which are revealed by literature research to be related to the estrogen-dependent signaling pathway, such as cell growth (amphiregulin, EGFR) [72,73,74], apoptosis (p53, Bcl-XL, survivin) [75,76], metastasis development (Dkk1, galectin 3, EpCAM, cathepsins) [77,78,79,80], and tumor immunoregulation (IL6, IL8, CCL20) [81,82,83,84]. An interesting yet conflicting phenomenon was observed during the summary of the protein expression experiments. Several tumor growth-promoting proteins were more prominently expressed following treatment, yet both tamoxifen and T15 are cytotoxic on our cell lines. This observation can be explained by the estrogen receptor expression profile of our cell lines. It is a well-described phenomenon that GPER1 signaling promotes tumor progression [85,86], helps cancer cells to obtain stem cell-like properties [87], and can mediate therapy resistance [88,89]. The presence of GPER1 poses a clinical problem as well, as patients with GPER1 expressing tumors have a worse survival rate, compared to patients with non-GPER1 expressing tumors, when receiving tamoxifen therapy [90,91,92], mostly because tamoxifen was described to be an agonist of GPER1 [36,37,38], inducing its previously mentioned tumor-promoting capability. 

GPER1 was seen to be present on all tested cell lines. However, the presence of the nuclear ERα and ERβ can counter the negative effects of the GPER1. On MCF7 and MDA-MB-231 cells, a fine balance between the ERα/ERβ inhibition and the activation of the prominently expressed GPER1 signaling was observed in the protein profile. Tamoxifen acts through arresting the cell cycle (S phase in MCF7 and G1 phase in MDA-MB-231), ultimately inhibiting tumor growth due to the inhibition of the ERα/ ERβ. However, the activation of GPER1 hinders its anti-tumor capability by raising the intracellular Amphiregulin, IL6, IL8, antiapoptotic proteins, and invasivity promoting factors. This effect of GPER1 was the most well observable on PANC1 cells, where only a very low level of ERα was detectable with a dominant presence of GPER1. Summarizing our result, the improved cytotoxic effect of T15 comes from the combination of its interaction with the ERs, namely the inhibition of the ERα/ERβ, which results in a cell cycle arrest (G1 phase, both MCF7 and MDA-MB231) and the ER-independent mechanism of the elevated ROS production due to the ferrocene group. The protein expression results upon T15 treatment suggest that this molecule is a weaker agonist of the GPER1 as it affects less of its target genes. 

It is important to mention that T15 has a beneficial anti-tumor effect even in racemic form, separating such enantiomers is still limited, however, our future experimental plans include the separate analysis of the two enantiomers. 

In conclusion, we were able to demonstrate two ferrocene-linked tamoxifen derivatives, with superior cytotoxicity compared to its parent molecule, capable of countering the effect of the commonly found GPER1, making them a possible candidate for further testing for possible clinical use.

## 4. Materials and Methods

### 4.1. Synthesis of the Investigated Tamoxifen Derivatives

The investigated T15 compound, a planar chiral ferrocene model [93], with structural characteristics closely related to that of tamoxifen, was prepared as a racemate (*R*_p_,*S*_p_)-1 and, for the sake of simplicity, (*R*_p_,*S*_p_)-T15 will be referred as 1 in Figure 8. This product was obtained from the racemic ferrocenocyclohexenone (*R*_p_,*S*_p_)-3 [94] in a 2-steps pathway comprising McMurry-type reductive coupling with 4,4′-dimethoxybenzophenone 5 effected by Zn/TiCl_4_ system and the subsequent BBr_3_-mediated *O*-demethylation of the racemic intermediate (*R*_p_,*S*_p_)-6. Employing the same synthetic strategy for the analogous conversion of propionylferrocene 4, the flexible diphenol 2 (T5 compound is referred as 2 in Figure 8) [95,96] with freely rotating ferrocenyl group, was also synthesized to be utilized as a positive control to reveal the impact of structural rigidity on the anti-tumor effect. For the experiments, the tamoxifen and its derivatives were solved in DMSO to prepare the stock solution.

### 4.2. Cell Lines and Culturing

The effects of tamoxifen and its ferrocene-linked derivatives, T15 and T5 were investigated on 3 cell lines, which were obtained from the European Collection of Authenticated Cell Cultures (ECACC, Salisbury, UK). All cell cultures were maintained at 37 °C in an incubator with a humidified atmosphere of 5% CO_2_. In every case, the base medium was supplemented with 10% FCS (Lonza Group Ltd., Basel, Switzerland), L-glutamine (2 mM), and 100 μg/mL penicillin/streptomycin (Gibco^®^/Invitrogen Corporation, New York, NY, USA). Mycoplasma testing was performed regularly on all cell lines.

PANC1 (87092802 ECCAC) is a pancreatic adenocarcinoma cell line established from a pancreatic carcinoma of ductal origin from a 56-years old Caucasian male. This cell culture was maintained in supplemented Dulbecco’s Modified Eagle Medium (Lonza Group Ltd., Basel, Switzerland).

MCF7 (86012803 ECCAC) is an ER-positive breast adenocarcinoma cell line established from the pleural effusion from a 69-years old Caucasian female. MCF7 cells were cultured in supplemented Minimum Essential Medium Eagle (Sigma-Aldrich, St. Louis, MO, USA) also containing 1% Non-Essential Amino Acid (Sigma-Aldrich, St. Louis, MO, USA).

MDA-MB-231 (92020424 ECCAC) is an ER-negative breast adenocarcinoma cell line also established from pleural effusion. Culturing of this cell line was conducted with supplemented Dulbecco’s Modified Eagle Medium (Lonza Group Ltd., Basel, Switzerland). 

### 4.3. Cytotoxicity Assays

The xCELLigence SP system (Roche Applied Science, Indianapolis, IN, USA) is an impedance-based method, that allows real-time monitoring of different cell biological properties, such as a quantitative readout of cell number, proliferation rate, cell size, and shape, and cell-substrate attachment quality. Adherent cells, such as PANC1 and MCF7, attach to the bottom of the wells and disrupt the electric current between the microelectrodes found on the bottom, which generates an increased impedance signal higher than the background signal of the system. Upon treating the cells with a cytotoxic compound, the cells acquire a round shape and detach from the bottom, which restores the electric current between the microelectrodes, and the measured impedance value decreases. A relative and dimensionless parameter, Cell index (*CI*), can be calculated from the detected impedance change by the following formula: *CI =(Z_i_ − Z_*0*_)/F* where *Z_i_* is the impedance at a given time point, *Z*_0_ is the impedance at t = 0 h, and *F_i_* is a constant depending on the frequency (*F*_10 kHz_ = 15). In order to allow real-time measurement, the system was placed in an incubator at 37 °C with a humidified atmosphere of 5% CO_2_. First, a background measurement was performed to acquire a baseline impedance curve by adding 100 μL of complete culture medium to each well and recording the *CI* for 1 h. Next, the cells were added to the wells at a concentration of 10,000 cells per well and cultured for 24 h. Upon completion of 24 h incubation, cells were treated with tamoxifen and its derivatives. *CI* values were measured for a further 72 h. Identical points of the concentration course study referred to the average of 3 parallel measurements. For comparison of the anti-tumor activity, the IC_50_ values were calculated from the *CI* values of each concentration obtained at 72 h with the RTCA 2.0 software (ACEA Biosciences, San Diego, CA, USA).

The viability of the non-adherent MDA-MB-231 cells treated with the tested compounds was determined by AlamarBlue assay (Thermo Fisher Scientific, Watham, MA, USA) according to the manufacturer’s instruction at 24, 48, and 72 h following treatment. After 4 h of incubation with the fluorescent reagent, the fluorescence intensity was measured with LS-50B Luminescence Spectrometer (Perkin Elmer Ltd., Buckinghamshire, UK) with the following settings: excitation wavelength = 560 nm and emission wavelength = 590 nm. Each data represents the average of 3 parallel measurements. IC_50_ values were calculated at 24, 48, and 72 h following treatment. 

To determine the IC_50_ value of derivatives, a dilution series of 9 concentrations (250 nM–100 µM) was made from the stock solution. Cell culture medium and DMSO (<1 vol%) were used as controls.

### 4.4. Cell Cycle Analysis

The effect of tamoxifen and its derivatives on the cell cycle was monitored by flow cytometry with PI incorporation. The stoichiometric binding of PI to double-stranded DNA allows the quantitative determination of the DNA content of cells and cell cycle analysis. Flow cytometric measurements were performed using the FACSCalibur flow cytometer (Becton Dickinson, San Jose, CA, USA). The sample preparation and the flow cytometric analysis bear a close resemblance to the description in [97]. The cells were seeded in 12-well plates at a concentration of 250,000 cells/well. After 24 h of incubation, they were treated with the derivative of choice in a concentration of 25 µM, as this was the EC_80_ value for tamoxifen at 72 h (i.e., the concentration where 80% of the cells are viable following treatment) on PANC1 cells. Cells were then harvested after 24 and 48 h of treatment and were fixed in ice-cold 70% ethanol and kept at −20 °C for 24 h. 

The samples were centrifuged and resuspended in RNase (100 μg/mL, Sigma-Aldrich, St. Louis, MO, USA) containing citric acid/sodium phosphate buffer (pH 7.8). Propidium iodide was added to the sample in 10 μg/mL concentration right before the flow cytometric measurements. PI fluorescence was collected with a 575/25 nm bandpass filter (FL3 channel) after linear amplification. For aggregate and debris discrimination, FL2-Width vs FL2-Area plot was used, and the gated cells were displayed in the FL2-Area histogram to assign percentage values to each population of cell cycle stages. For each measurement 25,000, single cells were collected. Data were analyzed by CellQuest Pro (Becton Dickinson, San Jose, CA, USA) and Flowing 2.5.1 (Turku Centre of Biotechnology, Turku, Finland) software. Statistical analysis was conducted with MS Excel and OriginPro 2018 (OriginLab Corporation, Northampton, MA, USA).

### 4.5. ROSGlo Assay

Cells were seeded in a white-walled, clear-bottom 96-well plate at a concentration of 10,000 cells/well, in 70 μL culture medium. After 24 h of incubation at previously described conditions, each well was treated with 10 μL of tamoxifen or T15 in a final concentration of 0.5, 2.5, and 25 μM. Culture medium and DMSO (<1 vol%) were used as negative controls, and 600 nM of bortezomib (bort.) was used as a positive control due to its well-known oxidative stress-inducing property [98,99,100]. Each data represents the average of 3 parallel measurements. After 24 h incubation with the compounds, components of the ROSGlo assay (Promega, Southhampton, UK) were prepared and added to the cells according to the manufacturer’s instructions. Luminescence was measured 24 h upon treatment with Fluoroskan^TM^ FL Microplate Fluorometer and Luminometer (Thermo Scientific, Waltham, MA USA).

### 4.6. Proteome Profiler Human Oncology XL Array

Cells were seeded to Petri dishes in a concentration of 1.5 million cells/dish, in 10 mL of culture medium. After 24 h of incubation at the previously described conditions, cells were treated with 25 μM tamoxifen or T15. Culture medium and DMSO (<1 vol%) were used as controls. 24 h upon treatment, cells were transferred to Eppendorf tubes and were washed with phosphate-buffered saline (PBS) and centrifuged at 1000× *g* at room temperature for 5 min. Lysis Buffer 17 (R&D Systems, Minneapolis, MN, USA) was added to each tube, and after 20 min of incubation, tubes were centrifuged at 20,000× *g* at 4 °C for 20 min. The supernatant containing the isolated proteins was kept as aliquots at −70 °C until use. Protein concentration was measured using the Pierce™ BCA Protein Assay Kit (Thermo Fisher Scientific, Watham, MA, USA). Each Proteome Profiler Human Oncology XL Array membrane (R&D Systems) was prepared with 200 μg of protein according to the manufacturer’s instructions. Membranes were exposed for 20 min to X-rays before reading chemiluminescence in the ChemiDoc XRS+ system (BioRad, Hercules, CA, USA). Image data were analyzed with the ImageLab 6.0.1 software (BioRad) as follows. Each data represents an average of 2 parallel samples represented by 2 spots for the same protein on the membrane. First, lanes were drawn around each column of the proteome profile membranes, and the spots were automatically detected as bands by the software in each lane. If for one protein, only 1 spot was detected by the software instead of 2, that protein was excluded from further processing. Secondly, every automatically detected protein-spot pair was manually circled with the same diameter in the software. The pixel intensity was measured in the circled area. The obtained mean pixel density was normalized with the mean of the reference spots for each membrane separately. A protein was recognized as an expressed one if its normalized pixel intensity reached 20% of the mean pixel intensity of the reference spots.

### 4.7. Detection of G-Protein Coupled Estrogen Receptor 1 (GPER1), Estrogen Receptor α (ERα) and Estrogen Receptor β (ERβ) Expression with Indirect Immunohistochemistry

Cells were seeded in a black wall, clear-bottom 96-well plate at a concentration of 10,000 cells/well, in 200 μL culture medium. After 24 h of incubation at the previously described conditions, cells were washed with PBS and fixed with 4% PFA at room temperature for 20 min. Next, one set of the samples was permeabilized with 0.5% Triton X Buffer. Anti-GPER1 (Biorbyt, St Louis, MO, USA, Catalogue No.: orb10740), anti-ERα (Thermo Fisher Scientific, Catalogue No.: PA1-308) and anti-Erβ (Thermo Fisher Scientific, Catalogue No.: MA524807) primary antibodies were added overnight. Secondary antibodies (ATTO 647N, Sigma Aldrich, Merck KgaA, Darmstadt, Germany, Catalogue No.: 40839, and Alexa Fluor 488, Thermo Fisher Scientific, Catalogue No.: A32731) was added 18 h later. Nucleus staining was conducted with Hoechst 33342 (Thermo Fisher Scientific, Catalogue No.: H3570). Secondary antibody without the primary was used as control. The samples were imaged by Celldiscoverer 7 system using 50 × Plan-Apochromat λ/0.35 NA water immersion objective with 2 × tube lens (Carl Zeiss AG, Jena, Germany) using the Z-stack function (minimum 20 Z-stacks per image). Then, all images were deconvolved to enhance the signal-to-noise ratio using ZEN Blue 2.6 software (Carl Zeiss AG, Jena, Germany). The intensity values and the number of nuclei were determined by ImageJ software (NIH, Bethesda, MD, USA).

### 4.8. Statistical Analysis

For statistical analysis, Origin Pro 8.0 (OriginLab Corporation, Northampton, MA, USA) and MS Excel software were used. One-way analysis of variance (ANOVA) followed by Fisher’s Least Significant Difference (Fisher’s LSD) post hoc test was performed. *p*-value less than 0.05 was considered statistically significant (*p* < 0.05: *, *p* < 0.01: **, *p* < 0.001: ***). The experiments were performed in triplicates (*n* = 3), and the data were presented as mean ± standard deviation (SD). 

## 5. Conclusions

The added ferrocene moiety increases the cytotoxic effect of tamoxifen on breast and pancreatic cancer cell lines. T5, T15, and tamoxifen are all capable of arresting the cell cycle, but in a concentration-dependent manner, i.e., cell cycle arrest was dominant in concentrations lower than the IC_50_ value of the compound. 

Additional research revealed several possible mechanisms that play a role in the cytotoxic effect of the novel tamoxifen derivatives. The dominant pathway they act through depends on the estrogen expression profile of the tumor. On MCF7 cells, where all three isoforms of the ER are expressed, the antagonistic effect on the ER α is predominant. However, GPER1 expression holds back the complete effectivity of tamoxifen, as it increases the expression levels of several tumor protective proteins. On MDA-MB-231 cells, only ER β is expressed in significant density. The antagonistic effect of tamoxifen on this isoform is clearly dominant on this cell line. On PANC1 cells, only GPER1 expression is detectable. The observed cytotoxic effects elicited by tamoxifen are explainable by non-estrogen receptor-dependent mechanisms, but the GPER1-induced tumor protective pathways interfere with these off-target effects. The ferrocene-linked novel tamoxifen derivatives T5 and T15 can counter the GPER1 induced tumor-promoting effects through direct cytotoxicity due to oxidative stress.

## Figures and Tables

**Figure 1 pharmaceuticals-15-00314-f001:**
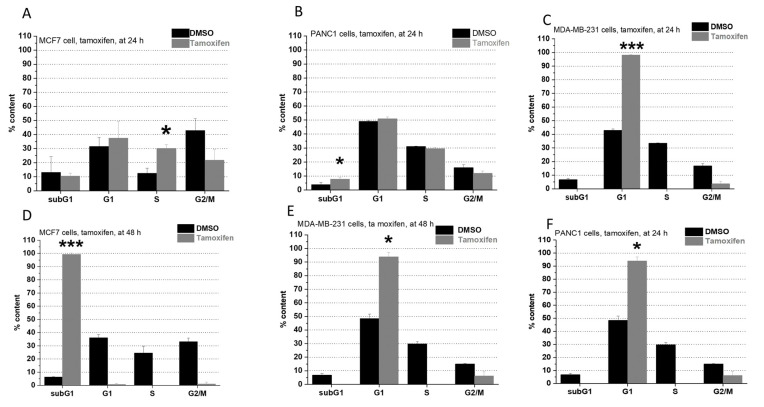
Impact of tamoxifen on the cell cycle on MCF7 ((**A**): at 24 h, (**D**): at 48 h), MDA-MB-231 ((**C**): at 24 h, (**E**): at 48 h), and PANC1 cells ((**B**): at 24 h, (**F**): at 48 h). Each cell line was treated with 25 µM. Each result represents the measurement of two parallel samples (*n* = 2). Data are given as mean values  ±  standard deviation (SD). Asterisks mark the subpopulation, where the number of cells is significantly higher compared to the DMSO control, and a cell cycle arrest is seen (*p* < 0.05: *, *p* < 0.001: ***) determined by the One-way ANOVA test followed by Fisher’s LSD post hoc test.

**Figure 2 pharmaceuticals-15-00314-f002:**
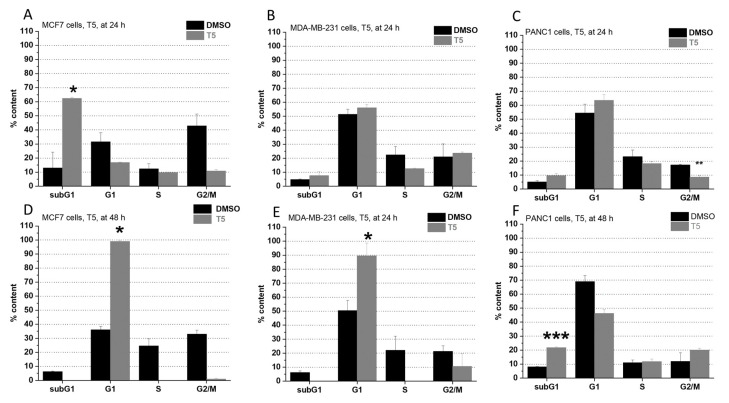
Impact of T5 on the cell cycle on MCF7 ((**A**): at 24 h, (**D**): at 48 h), MDA-MB-231 ((**B**): at 24 h, (**E**): at 48 h), and PANC1 cells ((**C**): at 24 h, (**F**): at 48 h). Each cell line was treated with 25 µM. Each result represents the measurement of two parallel samples (*n* = 2). Data are given as mean values  ±  standard deviation (SD). Asterisks mark the subpopulation, where the number of cells is significantly higher compared to the DMSO control, and a cell cycle arrest is seen (*p* < 0.05: *, *p* < 0.01: **, *p* < 0.001: ***) determined by the One-way ANOVA test followed by Fisher’s LSD post hoc test.

**Figure 3 pharmaceuticals-15-00314-f003:**
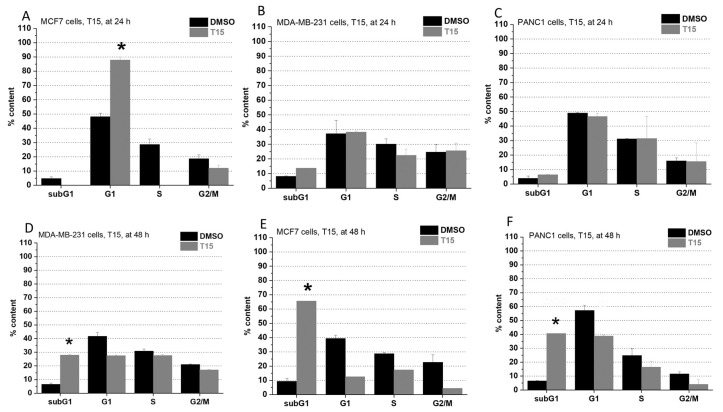
Impact of T15 on the cell cycle on MCF7 ((**A**): at 24 h, (**D**): at 48 h), MDA-MB-231 ((**B**): at 24 h, (**E**): at 48 h), and PANC1 cells ((**C**): at 24 h, (**F**): at 48 h). Each cell line was treated with 25 µM. Each result represents the measurement of two parallel samples (*n* = 2). Data are given as mean values ± standard deviation (SD). Asterisks mark the subpopulation, where the number of cells is significantly higher compared to the DMSO control, and a cell cycle arrest is seen (*p* < 0.05: *) determined by the One-way ANOVA test followed by Fisher’s LSD post hoc test.

**Figure 4 pharmaceuticals-15-00314-f004:**
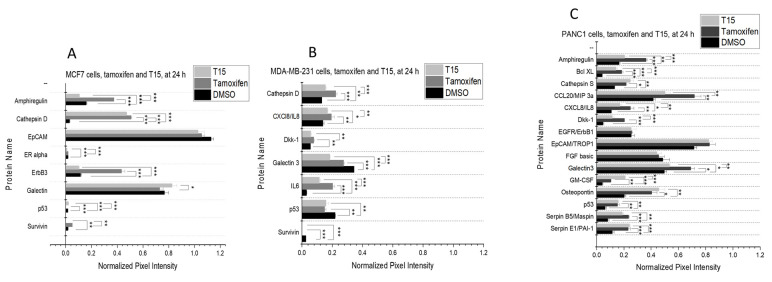
Protein expression changes of MCF7 (**A**), MDA-MB-231 (**B**), and PANC1 (**C**) cells after 24 h following treatment with tamoxifen and T15. Pictures of the original membranes (**A**) were taken after 10 min of exposition. Each protein pair, where at least one of the samples showed 20% of the pixel intensity of the reference points, were annotated then graphically represented (**B**). During the experiment, two parallel samples were used (*n* = 2). Data are given as mean values ± standard deviation (SD). Asterisks mark the significant differences between groups (*p* < 0.05: *, *p* < 0.01: **, *p* < 0.001: ***) determined by the One-way ANOVA test followed by Fisher’s LSD post hoc test.

**Figure 5 pharmaceuticals-15-00314-f005:**
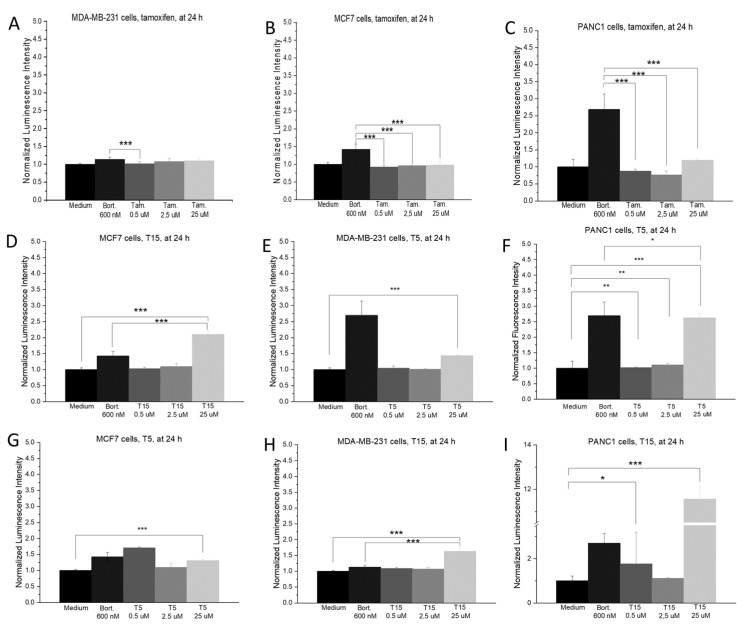
Impact of tamoxifen (**A**): MCF7, (**B**): MDA-MB-231, (**C**): PANC1), T5 (**D**): MCF7, (**E**): MDA-MB-231, (**F**): PANC1) and T15 ((**G**): MCF7, (**H**): MDA-MB-231, (**I**): PANC1) on the ROS production of the model cells. Each cell line was treated with 0.5, 2.5, and 25 µM tamoxifen derivatives, and 600 nM bortezomib (Bort.) was chosen as a positive control due to its well-known oxidative stress-increasing property. Each column on the graph represents the measurement of three parallel samples (*n* = 3). Data are given as mean values ± standard deviation (SD). Asterisks mark the concentrations, where the ROS production was significantly higher compared to the untreated, negative, medium control (*p* < 0.05: *, *p* < 0.01: **, *p* < 0.001: ***) determined by the One-way ANOVA test followed by Fisher’s LSD post hoc test.

**Figure 6 pharmaceuticals-15-00314-f006:**
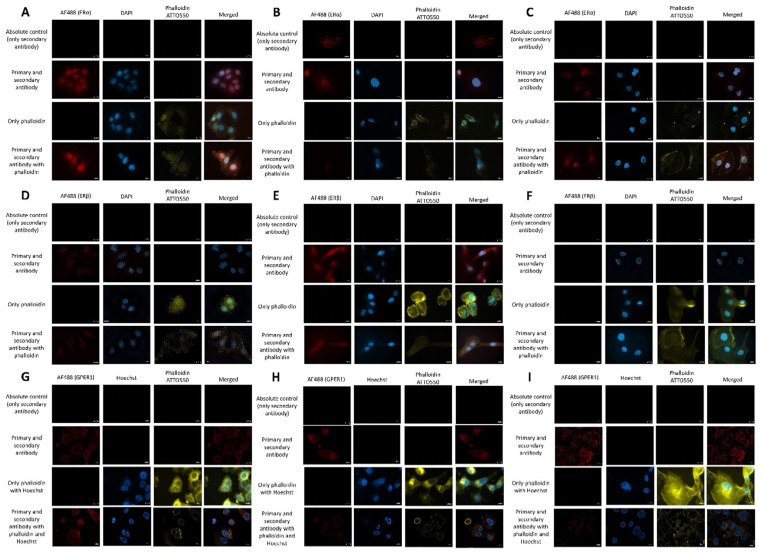
ERα ((**A**)-MCF7, (**B**)-MDA-MB-231, (**C**)-PANC1), ERβ ((**D**)-MCF7, (**E**)-MDA-MB-231, (**F**)-PANC1) and GPER1 expression ((**G**)-MCF7, (**H**)-MDA-MB-231, (**I**)-PANC1) of our model cells. Images were captured by Celldiscoverer 7 after Triton lysis. Magnification is at 100 ×, the scale bar is 10 µm.

**Figure 7 pharmaceuticals-15-00314-f007:**
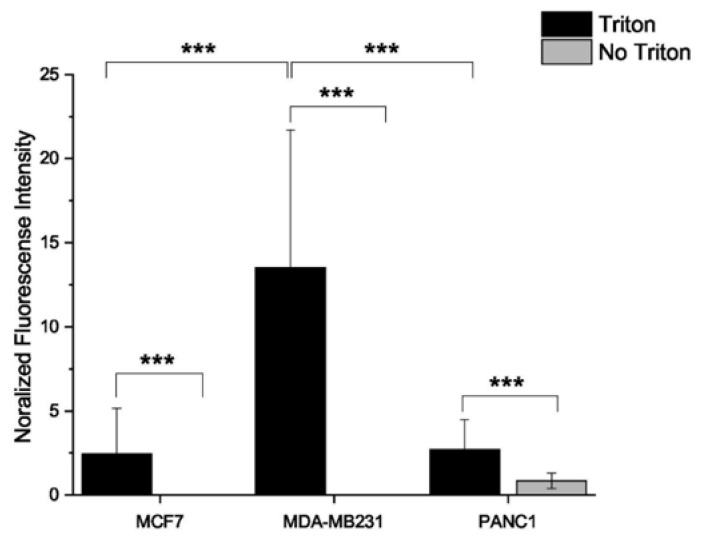
Quantification of the GPER1 expression. A total of 10 individual cells were chosen (*n* = 10) from images with merged channels of the receptor signal and the signal of the nuclear staining. After color channel splitting, the image thresholds were adjusted until the disappearance of the background signal, and the signal intensity was quantified with the same threshold in every picture. After the subtraction of the mean background signal, the mean fluorescence intensity was normalized by the number of nuclei. Data are given as mean values ± standard deviation (SD). Asterisks mark the significant differences between groups (*p* < 0.001: ***) determined by the One-way ANOVA test followed by Fisher’s LSD post hoc test.

**Figure 8 pharmaceuticals-15-00314-f008:**
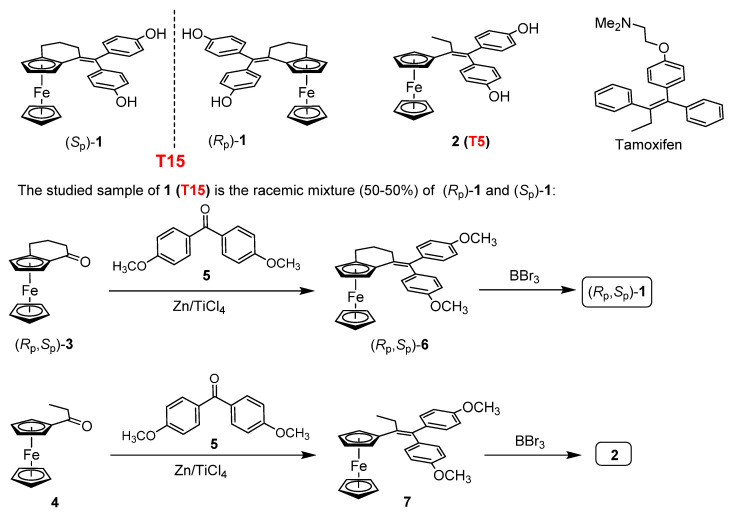
Chemical structure and the synthesis of the studied molecules.

**Table 1 pharmaceuticals-15-00314-t001:** Collection of different reports on ferrocifens.

Author	Reference	Name(s) of Derivative(s)	IC50 in µM
Kalabay M. et al.	-	T5 T15	43.34 (MCF7) 20.2 (MDA-MB-231) 12.46 (PANC1) 23.07 (MCF7), 23.77 (MDA-MB-231), 15.03 (PANC1)
Lainé A.-L.	[12]	P5, P15	0.5–2
Wang Y. et al.	[13]	FC1 FC2 FC3	1.5 (MDA-MB-231) 0.6 (MDA-MB-231) 0.5 (MDA-MB-231)
Cázares-Marinero, J. de J et al.	[14]	FcTAM, FcTAM-SAHA	2.6 (MDA-MB-231), 4.4 (MCF7) 0.7 (MDA-MB-231) 2 (MCF7)
Resnier P. et al.	[15]	ansa-FcDiOH FcDiOH	1 (SKMel28) 3 (SKMel28)
Topin-Ruiz S. et al.	[16]	p722	168 B16F10
Lu C. et al.	[17]	FcDiOH FcOHTam ansa-Fc-DiOH	0.6 (MDA-MB-231) 0.089 (MDA-MB-231) 0.5 (MDA-MB-231)
Cázares-Marinero, J. de J et al.	[18]	FcOHTam FcOHTam-SAHA FcOHTam-PSA FcOHTam-OPOA	1.6 (MDA-MB-231) 1.3 (MDA-MB-231) 1.5 (MCF7) 2.1 (MDA-MB-231) 4.4 (MCF7) 4.5 (MDA-MB-231) 6.6 (MCF7)
Buriez O. et al.	[19]	1 (ferrocenyl diphenol)	0.7 (MCF7) 0.5 (MDA-MB-231)
Wang Y. et al.	[20]	1a 2	0.66 (MDA-MB-231) 0.11 (MDA-MB-231)
Cunningham L. et al.	[21]	7 different racemic ferrocifens	09–16 (MCF7) 05–16.1 (MDA-MB-231)
Vessiéres A. et al.	[22]	FcOHTam	Fc-OH-TAM (0.1 and 1 uM) inhibited the growth of breast cancer cell lines MCF7, T-47D, ZR-75-1, MDA-MB-231, SKBR-3 and Hs578-T

**Table 2 pharmaceuticals-15-00314-t002:** IC_50_ values of tamoxifen, T5 and T15 on MCF7, MDA-MB-231, and PANC1 (C) cells. Each cell line was treated with concentrations between 250 nM and 25 μM. With each concentration, three parallel samples of the same cells were measured (*n* = 3). ND = non-detectable.

	MCF7			MDA-MB-231			PANC1		
	Tamoxifen	T5	T15	Tamoxifen	T5	T15	Tamoxifen	T5	T15
24 h	ND	42.81	24.12	25.53	29.97	25.28	33.33	21.17	24.46
48 h	42.73	42.95	23.24	23.05	31.6	23.6	33.68	21.29	23.80
72 h	42.71	43.34	23.07	21.81	26.3	23.77	33.79	12.46	15.03
96 h	44.35	42.94	12.71	not measured	not measured	not measured	46.13	non-detectable	5689

## Data Availability

The data generated and analyzed during our research are not available in any public database or repository but will be shared by the corresponding author upon reasonable request.

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
