# Peer review of "Investigation of the Antitumor Effects of Tamoxifen and Its Ferrocene-Linked Derivatives on Pancreatic and Breast Cancer Cell Lines"

_pharmaceuticals, 2022, doi:10.3390/ph15030314_

Round 1
Reviewer 1 Report
he article by Márton Kalabay et al. was written in the current field of search and development of new highly effective anticancer drugs based on conjugates of known anticancer drugs with functional fragments. potentially increasing cytotoxic activity, which in this study were ferrocene derivatives. The investigational drugs were synthesized and characterized, and further studied in various biological tests, such as cytotoxicity studies, cell cycle analysis, protein expression, estrogen receptor expression, ROS generation. The added ferrocene moiety has been shown to increase the cytotoxic effect of tamoxifen on breast and
pancreatic cancer cell lines. Possible mechanisms have also been proposed that play a role in the cytotoxic effect of the novel tamoxifen derivatives.
The article may be recommended for publication in its present form.
Author Response
Response to Review of manuscript
Response Reviewer 1
Manuscript ID: pharmaceuticals-1595087 – Major revision
Type of manuscript: Article
Title: Investigation of the antitumor effects of tamoxifen and its ferrocene-linked derivatives on pancreatic and breast cancer cell lines
Authors: Márton Kalabay, Zsófia Szász, Orsolya Láng, Eszter Lajkó, Éva
Pállinger, Cintia Duró, Tamás Jernei, Antal Csámpai, Angéla Takács,
László KÅ‘hidai *
Ms. Sukpattaraporn Ruangklai
Assistant Editor
Dear Editors and Reviewers,
We deeply appreciate Referees’ comments, corrections and most importantly the time they have spent on judging our manuscript. We have made all efforts to address all concerns that have been raised. Below you can find a point-by-point listing of our replies. The total manuscript with all major and minor changes is also attached/uploaded to facilitate the work of Editors and Referees. All changes in the manuscript were tracked with the “Track changes” feature of MS Word and marked up by adding a comment as well.
As practically all the suggested modifications of the Referees were accepted and we also paid attention to answering the questions raised, we find the revised version of the manuscript significantly improved and the Authors are grateful for the Referees’ help in getting this done.
Response to Reviewer 1
The Authors would like to thank Referee’s time and effort in correcting the manuscript, we are grateful for their kind comments and recommendation for acceptance in the current form.

Reviewer 2 Report
The research paper entitled " Investigation of the antitumor effects of tamoxifen and its fer-rocene-linked derivatives on pancreatic and breast cancer cell lines (Manuscript ID: pharmaceuticals-1595087)” was reviewed. After reading the manuscript, I suggest to author to revise wisely for publication in Pharmaceuticals. Please do the revision for this manuscript based on comments below (major revision):
- The abstract should be re-written to summarize the work; the abstract should state briefly the purpose of the research, the PRINCIPLE results and MAJOR conclusions. An abstract is often presented separately from the article, so it must be able to stand alone.
- The novelty of this study compared to other studies is not clear.
- The mechanism of cytotoxicity is not clear. Please explain schematically.
4 The format of figures in the manuscript is not uniform.
- Authors must compare the efficiency of this research (in a Table) with others reported in the literature.
- About results and discussion, explanation is poor, suitable and more analysis should be added. It is not acceptable at all in this format.
- The authors must revise the manuscript carefully to eliminate grammatical errors and typo-errors.
Author Response
Response to Review of manuscript
Response Reviewer 2
Manuscript ID: pharmaceuticals-1595087 – Major revision
Type of manuscript: Article
Title: Investigation of the antitumor effects of tamoxifen and its ferrocene-linked derivatives on pancreatic and breast cancer cell lines
Authors: Márton Kalabay, Zsófia Szász, Orsolya Láng, Eszter Lajkó, Éva
Pállinger, Cintia Duró, Tamás Jernei, Antal Csámpai, Angéla Takács,
László KÅ‘hidai *
Ms. Sukpattaraporn Ruangklai
Assistant Editor
Dear Editors and Reviewers,
We deeply appreciate Referees’ comments, corrections and most importantly the time they have spent on judging our manuscript. We have made all efforts to address all concerns that have been raised. Below you can find a point-by-point listing of our replies. The total manuscript with all major and minor changes is also attached/uploaded to facilitate the work of Editors and Referees. All changes in the manuscript were tracked with the “Track changes” feature of MS Word and marked up by adding a comment as well.
As practically all the suggested modifications of the Referees were accepted and we also paid attention to answering the questions raised, we find the revised version of the manuscript significantly improved and the Authors are grateful for the Referees’ help in getting this done.
Response to Reviewer 2
- The abstract should be re-written to summarize the work; the abstract should state briefly the purpose of the research, the PRINCIPLE results and MAJOR conclusions. An abstract is often presented separately from the article, so it must be able to stand alone.
Authors agree with Referee that the abstract could have been a better summary of our research, so we have rewritten it to be presentable alone for the following:
Tamoxifen is a long-known antitumor drug, which is the gold standard therapy in estrogen receptor (ER) positive breast cancer patients. According to previous studies, the conjugation of the original tamoxifen molecule with different functional groups can significantly improve its antitumor effect. The purpose of this research was to uncover the molecular mechanisms behind the cytotoxicity of different ferrocene-linked tamoxifen derivates. Tamoxifen and its ferrocene-linked derivatives, T5 and T15 were tested in PANC1, MCF7 and MDA-MB-231 cells, where the incorporation of the ferrocene group improved the cytotoxicity on all cell lines. PANC1, MCF7 and MDA-MB-231 express ERα and GPER1 (G-protein coupled ER 1), however, ERβ is only expressed by MCF7 and MDA-MB-231 cells. Tamoxifen is a known agonist of GPER1, a receptor that can promote tumor progression. Analysis of the protein expression profile showed that while being cytotoxic, tamoxifen elevated the levels of different tumor growth-promoting factors (e.g. Bcl-XL, Survivin, EGFR, Cathepsins, chemokines), on the other hand, the ferrocene-linked derivates were able to lower these proteins. Further analysis showed that the ferrocene-linked derivatives significantly elevate the cellular oxidative stress compared to tamoxifen treatment. In conclusion, we were able to find two molecules possessing better cytotoxicity compared to their unmodified parent molecule, while also being able to counter the negative effects of the presence of the GPER1 through the ER-independent mechanism of oxidative stress induction.
- The novelty of this study compared to other studies is not clear.
Tamoxifen is a long-known anticancer agent, and in the past two decades, several studies aimed to find new, possibly better derivates based on the original molecule. The novelty of this research lies in two aspects. Ferrocene-linked derivates of tamoxifen are not found commonly throughout the literature, apart from the work of Gerald Jaouen and his colleagues, who first synthesized the T5 molecule used in our work, are only a handful of authors who worked with similar derivates. Some articles report further modified versions of derivates, encapsulated in lipid nanoparticles, or conjugated with other functional groups. However, a small fraction of these research articles describe the molecular biological means of the action of the derivates covering a wide spectrum of possible pathways behind its mechanism, they mainly focus on the chemistry and synthesis of the derivates, and basic cytotoxicity measurements. Our research also deals with clinically relevant questions, such as the presence of GPER1 and its possible effect on the treatment efficiency on a molecular level. We also demonstrate that with these derivates the unlucky constellation of the estrogen receptor isoforms can be overcome in vitro and there is a need for further insight on whether they can be used in in vivo experiments featuring GPER1 expressing tumors.
- The mechanism of cytotoxicity is not clear. Please explain schematically.
The incorporation of the ferrocene group into the original tamoxifen molecule opens a new mechanism of action besides the existing SERM-like property of the compound. The pharmacophore region found both in tamoxifen and its two ferrocene-linked derivates is capable of binding to both the nuclear ERα and ERβ and the GPER1. Upon binding, the nuclear ER isoforms are inhibited in the breast and pancreatic tissue, leading to lower transcriptional activity and blocked tumor proliferation. However, the pharmacophore region of tamoxifen activates the GPER1, which acts as an oncogene, regulating the transcription of several factors involved in the survival of tumor cells such as growth factors, cytokines, apoptotic regulators. According to our results, there is a fine balance between the effect of the nuclear ERs and the GPER1. Tamoxifen acts through the inhibition of the nuclear ERs, however, its complete effect is partly countered by the activation of the GPER1. The ferrocene linked derivates are also capable of interacting with the ERs, but the oxidative stress inflicted by the ferrocene group is completely independent of the ER signaling pathway and can strengthen the antitumor effect caused by the inhibition of the nuclear ERs resulting in improved cytotoxicity compared to tamoxifen, even on GPER1 expressing tumor cells.
- The format of the figures in the manuscript is not uniform.
The Authors agree with Referee’s comment, so we changed the color, legend placement, font sizes to be uniform throughout our figures, except for Figures 6, 8 and S1-5.
- Authors must compare the efficiency of this research (in a Table) with others reported in the literature.
The Authors appreciate the Referee’s request for the need for comparison with different works found in the literature as it further broadened our knowledge on ferrocene-linked derivates. Literature searches Please find the table below with references to the mentioned works. We also included this table in the main article in the introduction section.
Author |
Reference |
Name(s) of derivative(s) |
IC50 in µM |
Kalabay M. et al. |
- |
T5 |
43.34 (MCF7) |
20.2 (MDA-MB-231) |
|||
12.46 (PANC1) |
|||
T15 |
23.07 (MCF7) |
||
23.77 (MDA-MB-231) |
|||
15.03 (PANC1) |
|||
Lainé A.-L. |
[12] |
P5, P15 |
0.5-2 |
Yong W et al. |
[13] |
FC1 |
1.5 (MDA-MB-231) |
FC2 |
0.6 (MDA-MB-231) |
||
FC3 |
0.5 (MDA-MB-231) |
||
Cázares-Marinero, J. de J et al. |
[14] |
FcTAM, |
2.6 (MDA-MB-231) |
FcTAM-SAHA |
4.4 (MCF7) |
||
Resnier P. et al. |
[15] |
ansa-FcDiOH |
1 (SKMel28); |
0,7 (MDA-MB-231); |
|||
2 (MCF7) |
|||
FcDiOH |
3 (SKMel28) |
||
Topin-Ruiz S. et al. |
[16] |
p722 |
168 B16F10 |
Lu C. et al. |
[17] |
FcDiOH |
0.6 (MDA-MB-231) |
FcOHTam |
0.089 (MDA-MB-231) |
||
ansa-Fc-DiOH |
0.5 (MDA-MB-231) |
||
Cázares-Marinero, J. de J et al. |
[18] |
FcOHTam |
1.6 (MDA-MB-231) |
FcOHTam-SAHA |
1.3 (MDA-MB-231) |
||
1.5 (MCF7) |
|||
FcOHTam-PSA |
2.1 (MDA-MB-231) |
||
4.4 (MCF7) |
|||
FcOHTam-OPOA |
4.5 (MDA-MB-231) |
||
6.6 (MCF7) |
|||
Buriez O. et al. |
[19] |
1 ferrocenyl diphenol |
0.7 (MCF7) |
0.5 (MDA-MB-231) |
|||
Wang Y. et al. |
[20] |
1a |
0.66 (MDA-MB-231) |
2 |
0.11 (MDA-MB-231) |
||
Cunningham L. et al. |
[21] |
7 different racemic ferrocifens |
09-16 (MCF7) |
05-16.1 (MDA-MB-231) |
|||
Vessiéres A. et al. |
[22] |
FcOHTam |
Fc-OH-TAM (0.1 and 1 uM) inhibited growth of breast cancer cell lines MCF7, T-47D, ZR-75-1, MDA-MB-231, SKBR-3 and Hs578-T |
- Lainé, A.L., E. Adriaenssens, A. Vessières, G. Jaouen, C. Corbet, E. Desruelles, P. Pigeon, R.A. Toillon, and C. Passirani, The in vivo performance of ferrocenyl tamoxifen lipid nanocapsules in xenografted triple negative breast cancer. Biomaterials, 2013. 34(28): p. 6949-56.
- Wang, Y., M.A. Richard, S. Top, P.M. Dansette, P. Pigeon, A. Vessières, D. Mansuy, and G. Jaouen, Ferrocenyl Quinone Methide-Thiol Adducts as New Antiproliferative Agents: Synthesis, Metabolic Formation from Ferrociphenols, and Oxidative Transformation. Angew Chem Int Ed Engl, 2016. 55(35): p. 10431-4.
- Cázares Marinero Jde, J., M. Lapierre, V. Cavaillès, R. Saint-Fort, A. Vessières, S. Top, and G. Jaouen, Efficient new constructs against triple negative breast cancer cells: synthesis and preliminary biological study of ferrocifen-SAHA hybrids and related species. Dalton Trans, 2013. 42(43): p. 15489-501.
- Resnier, P., N. Galopin, Y. Sibiril, A. Clavreul, J. Cayon, A. Briganti, P. Legras, A. Vessières, T. Montier, G. Jaouen, J.-P. Benoit, and C. Passirani, Efficient ferrocifen anticancer drug and Bcl-2 gene therapy using lipid nanocapsules on human melanoma xenograft in mouse. Pharmacological Research, 2017. 126: p. 54-65.
- Topin-Ruiz, S., A. Mellinger, E. Lepeltier, C. Bourreau, J. Fouillet, J. Riou, G. Jaouen, L. Martin, C. Passirani, and N. Clere, p722 ferrocifen loaded lipid nanocapsules improve survival of murine xenografted-melanoma via a potentiation of apoptosis and an activation of CD8(+) T lymphocytes. Int J Pharm, 2021. 593: p. 120111.
- Lu, C., J.M. Heldt, M. Guille-Collignon, F. Lemaître, G. Jaouen, A. Vessières, and C. Amatore, Quantitative analyses of ROS and RNS production in breast cancer cell lines incubated with ferrocifens. ChemMedChem, 2014. 9(6): p. 1286-93.
- Cázares-Marinero Jde, J., S. Top, A. Vessières, and G. Jaouen, Synthesis and antiproliferative activity of hydroxyferrocifen hybrids against triple-negative breast cancer cells. Dalton Trans, 2014. 43(2): p. 817-30.
- Buriez, O., J.M. Heldt, E. Labbé, A. Vessières, G. Jaouen, and C. Amatore, Reactivity and antiproliferative activity of ferrocenyl-tamoxifen adducts with cyclodextrins against hormone-independent breast-cancer cell lines. Chemistry, 2008. 14(27): p. 8195-203.
- Wang, Y., F. Heinemann, S. Top, A. Dazzi, C. Policar, L. Henry, F. Lambert, G. Jaouen, M. Salmain, and A. Vessieres, Ferrocifens labelled with an infrared rhenium tricarbonyl tag: synthesis, antiproliferative activity, quantification and nano IR mapping in cancer cells. Dalton Trans, 2018. 47(29): p. 9824-9833.
- Cunningham, L., Y. Wang, C. Nottingham, J. Pagsulingan, G. Jaouen, M.J. McGlinchey, and P.J. Guiry, Enantioselective Synthesis of Planar Chiral Ferrocifens that Show Chiral Discrimination in Antiproliferative Activity on Breast Cancer Cells. Chembiochem, 2020. 21(20): p. 2974-2981.
- Vessières, A., C. Corbet, J.M. Heldt, N. Lories, N. Jouy, I. Laïos, G. Leclercq, G. Jaouen, and R.A. Toillon, A ferrocenyl derivative of hydroxytamoxifen elicits an estrogen receptor-independent mechanism of action in breast cancer cell lines. J Inorg Biochem, 2010. 104(5): p. 503-11.
- About Results and Discussion, the explanation is poor, suitable and more analysis should be added. It is not acceptable at all in this format.
Results and discussion were rewritten to be more focused on the three steps (cell cycle measurements, ROS quantification and protein expression profile analysis) done to screen for the possible background mechanism of the improved cytotoxicity of the derivatives and provide a conclusion summarizing the interaction between the ER-dependent and off-target pathways.
- The Authors must revise the manuscript carefully to eliminate grammatical errors and typo errors.
The Authors apologize for the grammatical errors. The manuscript and the supporting information files were carefully checked to correct the linguistic inaccuracies.
Reviewer 3 Report
The authors have investigated the anticancer role of ferrocene-linked tamoxifen compounds (T5 and T15) comparing their toxicity mechanisms with those of well-known tamoxifen. The in vitro evaluation was performed in one pancreatic adenocarcinoma (PANC1) and two breast adenocarcinoma cell lines (MCF7 and MDA-MB-231).
Overall, the methodologies used in this study are appropriate for the aim of the study. The results are interesting and well discussed despite the figure numbers and references in the text are incorrect. For instance, the supplementary figures are not cited in main text. Please correct them accordingly. Furthermore, I suggest some minor revisions that may improve the quality of the work:
- In section 2.1, the IC50 value at 72h of T5 on MDA-MB-231 cells is different between the text and table 1 (20.2 mM in the text vs 26.3 mM in table 1).
- In section 2.4, please correct the reference “(Figure 5D-E)”.
- Please add the extended name of all proteins when they are first written (i.e. Epidermal Growth Factor Receptor (EGFR)).
- In Figures S1, S2 and S3, please add more details (i.e. add cell line and treatment on each dose-response curve and the significant differences between the curves).
- In all figures please add titles of cell line and drugs when it is required (more drugs, cell line types, timing).
Author Response
Response to Review of manuscript
Response Reviewer 3
Manuscript ID: pharmaceuticals-1595087 – Major revision
Type of manuscript: Article
Title: Investigation of the antitumor effects of tamoxifen and its ferrocene-linked derivatives on pancreatic and breast cancer cell lines
Authors: Márton Kalabay, Zsófia Szász, Orsolya Láng, Eszter Lajkó, Éva
Pállinger, Cintia Duró, Tamás Jernei, Antal Csámpai, Angéla Takács,
László KÅ‘hidai *
Ms. Sukpattaraporn Ruangklai
Assistant Editor
Dear Editors and Reviewers,
We deeply appreciate Referees’ comments, corrections and most importantly the time they have spent on judging our manuscript. We have made all efforts to address all concerns that have been raised. Below you can find a point-by-point listing of our replies. The total manuscript with all major and minor changes is also attached/uploaded to facilitate the work of Editors and Referees. All changes in the manuscript were tracked with the “Track changes” feature of MS Word and marked up by adding a comment as well.
As practically all the suggested modifications of the Referees were accepted and we also paid attention to answering the questions raised, we find the revised version of the manuscript significantly improved and the Authors are grateful for the Referees’ help in getting this done.
Response to Reviewer 3
- In section 2.1, the IC50 value at 72h of T5 on MDA-MB-231 cells is different between the text and table 1 (20.2 mM in the text vs 26.3 mM in table 1).
The Authors apologize for the inaccuracy. The correct IC50 value is the one in Table 1, so we changed the one in the text.
1. In section 2.4, please correct the reference “(Figure 5D-E)”.
The Authors apologize for the inaccuracy; the reference was corrected.
2. Please add the extended name of all proteins when they are first written (i.e. Epidermal Growth Factor Receptor (EGFR)).
Extended names of proteins were added at their first appearance in the text.
3. In Figures S1, S2 and S3, please add more details (i.e. add cell line and treatment on each dose-response curve and the significant differences between the curves).
Cell line and treatment was added in the title of each graph. However, significance can not be marked on this kind of graph, as the y axis shows the normalized value of the Cell Index, and statistical probes require at least two values, in this case, we have only the normalized one. If the significance is a key requirement, we gladly redesign these graphs, but we think it is representative enough for the screening of the IC50 values.
4. In all figures, please add titles of cell lines and drugs when it is required (more drugs, cell line types, timing).
Cell line name and treatment was added to each graph title.
Round 2
Reviewer 2 Report
Accept